# Suppression of Streptococcosis and Modulation of the Gut Bacteriome in Nile Tilapia (*Oreochromis niloticus*) by the Marine Sediment Bacteria *Bacillus haynesii* and *Advenella mimigardefordensis*

Md Mahfuzur Rahman,[a] Sulav Indra Paul,[a] Ashikur Rahman,[a] Md Shameul Haque,[a] Md Ali Arman Ador,[a] Md Javed Foysal,[b] Md Tofazzal Islam,[a] Md Mahbubur Rahman[a]

[a]Institute of Biotechnology and Genetic Engineering, Bangabandhu Sheikh Mujibur Rahman Agricultural University, Gazipur, Bangladesh
[b]Department of Genetic Engineering and Biotechnology, Shahjalal University of Science and Technology, Sylhet, Bangladesh

Md Mahfuzur Rahman and Sulav Indra Paul contributed equally to this article. The order of co-first authors was determined alphabetically in the order of their first names.

**ABSTRACT** Streptococcosis is one of the major threats to Nile tilapia (*Oreochromis niloticus*) in most regions of the world. Recently, *Enterococcus faecalis* has been widely reported to be involved in streptococcosis in *O. niloticus* in Asia and Africa. This study aimed to isolate beneficial marine bacteria to evaluate their effects on growth, hematological parameters, nonspecific immunity, the gut bacteriome, and streptococcosis prevention efficacy in *O. niloticus*. A total of 36 marine soil bacteria were isolated, and *in vitro* screening was conducted to determine their antibacterial activities against fish pathogens. Two antagonistic bacteria were identified based on 16S rRNA gene sequencing, *Bacillus haynesii* CD223 and *Advenella mimigardefordensis* SM421. These bacteria were incorporated into fish feed and fed to *O. niloticus* for 90 days. The application of these strains via incorporation into fish feed significantly promoted growth, improved hematological parameters and immunoglobulin M (IgM) levels, modulated the gut bacteriome by reducing the load of pathogenic *Enterococcus* spp., and developed disease prevention efficacy in *O. niloticus*. Furthermore, *in vivo* assays revealed that the inclusion of extracellular products (ECPs) (at 250 $\mu$g mL$^{-1}$) of CD223 and SM421 with feed significantly enhanced the rate of survival (100%) of *O. niloticus* from streptococcosis compared to the controls (only 30%). The ECPs of these bacteria also prevented 90 to 100% of fish from developing streptococcosis. These strains could be promising for safe use in *O. niloticus* farming to prevent and control the emergence of streptococcosis caused by *E. faecalis*.

**IMPORTANCE** Nile tilapia (*Oreochromis niloticus*) is one of the most economically important cultured fish species throughout the world. Streptococcosis is a significant threat to global Nile tilapia farming. *Enterococcus faecalis* has recently emerged as an important pathogen of streptococcosis in Asia and Africa. The application of antibiotics and probiotics and vaccination are the major ways to combat streptococcosis. However, the extensive use of antibiotics leads to the development of antibiotic resistance in pathogenic as well as environmental bacteria, which is a great threat to public health. There is no study on preventing streptococcosis caused by *E. faecalis* using beneficial bacteria. For the first time, the present study demonstrated that two marine bacteria, *Bacillus haynesii* strain CD223 and *Advenella mimigardefordensis* strain SM421, have great potential for controlling streptococcosis in Nile tilapia. These bacteria also enhanced the growth, improved hematological parameters and IgM levels, and positively modulated the gut bacteriome of Nile tilapia.

**KEYWORDS** *Bacillus haynesii*, *Advenella mimigardefordensis*, *Enterococcus faecalis*, streptococcosis, antimicrobial activity, disease prevention, *Oreochromis niloticus*

**Ad Hoc Peer Reviewer** Pabodha Weththasinghe; Uriel Rodriguez-Estrada

Address correspondence to Md Mahbubur Rahman, mahbub-biotech@bsmrau.edu.bd.

The authors declare no conflict of interest.

Aquaculture has progressively become a potential source of animal protein around the world, accounting for approximately 44% (74 million tons) of the total global fish production, with a value of $160 billion (1). The prodigious growth of aquaculture production has greatly contributed to the nutritional security of many developing countries (2, 3). However, the emergence and prevalence of various diseases have led to severe production and financial losses in the aquaculture sector (4). For the sustainability of this sector, proper disease management and biosecurity measures are imperative to reduce losses. Different therapeutic and prophylactic agents, including antibiotics, are used to control pathogens (5). However, the extensive use of antibiotics leads to the development of antibiotic resistance in pathogenic as well as environmental bacteria, posing a great threat to public health. Thus, the development of new eco-friendly agents to combat drug resistance in bacterial pathogens is urgently needed.

Saint Peter's fish, the Nile tilapia (*Oreochromis niloticus*), is one of the most commercially important aquaculture species throughout the world (6). This fish is reported to be susceptible to different bacterial diseases (streptococcosis, motile *Aeromonas* septicemia, and francisellosis, etc.) (7), viral infections (by tilapia lake virus [TiLV], nervous necrosis virus [NNV], and infectious pancreatic necrosis virus [IPNV], etc.) (8), and parasitic diseases (monogeniasis and trichodiniasis, etc.) (9). Among the bacterial diseases, streptococcosis is considered a major threat, causing substantial annual losses in tilapia production worldwide (10–12). *Streptococcus iniae*, *S. agalactiae*, *S. parauberis*, and *S. dysgalactiae* are recognized as the prominent causative agents of streptococcosis in different geographical regions (13–16). In addition, *Lactococcus garvieae*, *Vagococcus* sp., *Enterococcus faecalis*, *E. faecium*, and *E. hirae* have also been reported as notable causative agents of streptococcosis in Asian countries such as India (17), Bangladesh (10, 11, 18), Philippines (19), and Indonesia (20) as well as an African country, Egypt (21, 22). Enterococci were previously classified as streptococci of Lancefield group D (23). *Enterococcus* sp. has also been reported as a pathogen in many fish species such as Japanese yellow tail (*Seriola quinqueradiata*) (24), turbot (*Scophthalmus maximus*) (25), walking catfish (*Clarias batrachus*) (10), stinging catfish (*Heteropneustes fossilis*) (10), silver barb (*Barbonymus gonionotus*) (18), red tilapia (*Oreochromis* hybrid) (20), and flathead gray mullet (*Mugil cephalus*) (26).

The control of streptococcosis usually relies on the application of chemotherapeutic agents, vaccines, and environmental management strategies, but vaccination is usually less effective in preventing outbreaks of streptococcosis (27, 28). Disinfectants may be a good option for controlling the disease, but dissolved disinfectants may have hazardous side effects on aquatic ecosystems (28). In this respect, the oral administration of beneficial microbes or their derivatives could be an excellent disease control strategy to prevent streptococcosis in Nile tilapia.

Soil is a rich bioresource and promotes the growth of various types of microorganisms (29). Marine soil sediments harbor diverse microorganisms that are known to produce novel bioactive natural products with important biological activities such as antibacterial, antifungal, antiviral, antitumor, and anticancer activities (30). Moreover, soil-derived bacteria are prolific producers of diverse antimicrobial compounds that can inhibit the growth of many pathogens (31). Furthermore, bacterial species isolated from marine environments can improve the aquaculture environment and can be used as probiotics for *O. niloticus* (32–34). Saint Martin's Island in the Bay of Bengal region of Bangladesh is an untapped reservoir of diverse economically important beneficial microorganisms. In a recent study, we demonstrated that some marine sponge-associated bacteria of Saint Martin's Island in the Bay of Bengal significantly prevent motile *Aeromonas* septicemia in the cultured fish species *Labeo rohita* (35). Therefore, in this study, we aimed to discover novel strains of bacteria from the soil sediments of Saint Martin's Island with the potential to control streptococcosis in Nile tilapia. The specific objectives of this study were to (i) isolate and characterize antagonistic bacteria from the soil sediments of Saint Martin's Island against *E. faecalis*, which causes streptococcosis in Nile tilapia; (ii) assess the effects of the dietary inclusion of these bacteria and the extracts of their culture supernatants on

the growth, immune response, and gut microbial communities of Nile tilapia; and (iii) evaluate the efficacy of these bioagents in the prevention of *E. faecalis* infection in Nile tilapia.

## RESULTS

**Isolation and *in vitro* inhibition assay of marine soil bacterial isolates.** Thirty-six bacterial isolates were obtained from the soil sediments of Saint Martin's Island in the Bay of Bengal, Bangladesh. Four bacterial groups were identified based on the preliminary phenotypic characteristics of these isolates (Table 1). Further *in vitro* inhibition assays revealed that the cell-free culture supernatants of isolates CD223 and SM421 exhibited inhibitory activities against the fish-pathogenic *E. faecalis* strains FF11 and F1B1 (Table 2). Based on phenotypic characteristics (Table 1), isolate CD223 was preliminarily identified as *Bacillus* sp. by Paul et al. (35), and isolate SM421 was identified as *Advenella* sp. by Coenye et al. (36).

**Molecular identification of the selected marine bacterial isolates.** The 16S rRNA gene sequence of isolate CD223 exhibited 99.43% homology with that of *Bacillus haynesii* strain NRRL B-41327, and isolate SM421 showed 99.78% homology with *Advenella mimigardefordensis* strain DPN7. Phylogenetic tree analysis also supported the results of the homology study and confirmed the close positioning of *B. haynesii* strain CD223 and *A. mimigardefordensis* strain SM421 with the reference strains *B. haynesii* NRRL B-41327 and *A. mimigardefordensis* DPN7, respectively (Fig. 1).

**Bile esculin activity, tolerance to different pH levels, and cytotoxic effects of CD223 and SM421.** Both strains CD223 and SM421 exhibited hydrolytic activity toward bile esculin and could grow under both acidic and alkaline conditions (pH range of 3 to 9) (Fig. 2). The *in vivo* cytotoxic and toxigenic effects of the extracellular products (ECPs) of *B. haynesii* CD223 and *A. mimigardefordensis* SM421 on the survival of hatched brine shrimp (*Artemia salina*) nauplii (at developmental stage II) were assessed to measure their safety in animal hosts. The ECPs extracted from strains CD223 and SM421 did not show any significant cytotoxic or toxigenic effects on the survival of *A. salina* nauplii compared to the control.

**Growth performance of *O. niloticus* upon dietary inclusion of *B. haynesii* CD223 and *A. mimigardefordensis* SM421.** The growth performance of *O. niloticus* after dietary supplementation with *B. haynesii* CD223 and *A. mimigardefordensis* SM421 was found to be significantly ($P < 0.05$) better than that of the control. The body weight gains (percentages) obtained in the *B. haynesii* CD223- and *A. mimigardefordensis* SM421-fed fish and the control fish were 114.59% $\pm$ 3.84%, 221.68% $\pm$ 9.76%, and 48.28% $\pm$ 2.21%, respectively (Table 3). The specific growth rate (SGR) (percentage per day) values obtained for the *B. haynesii* CD223- and *A. mimigardefordensis* SM421-fed fish and the control fish were 1.23% $\pm$ 0.13%, 1.92% $\pm$ 0.11%, and 0.62% $\pm$ 0.12%, respectively (Table 3).

***B. haynesii* CD223 and *A. mimigardefordensis* SM421 improved hematological parameters and IgM levels in *O. niloticus*.** The hematological parameters (hemoglobin, white blood cell [WBC], and red blood cell [RBC] levels) and nonspecific immunity (immunoglobulin M [IgM] level) in *B. haynesii* CD223- and *A. mimigardefordensis* SM421-fed fish were found to be significantly ($P < 0.05$) higher than those of the controls (Table 3). Among all of the treatments, the highest levels of hemoglobin (6.17 $\pm$ 0.1 g dL$^{-1}$) and WBCs (3.65 $\times$ 10$^7$ $\pm$ 0.2 $\times$ 10$^7$ cells mL$^{-1}$) were found in CD223-fed fish compared to the controls. On the contrary, the highest levels of RBCs (2.83 $\times$ 10$^5$ $\pm$ 0.09 $\times$ 10$^5$ cells mL$^{-1}$) and IgM (108.33 $\pm$ 3.3 $\mu$g dL$^{-1}$) were found in SM421-fed fish compared to the controls (Table 3).

***B. haynesii* CD223 and *A. mimigardefordensis* SM421 modulated the gut bacterial flora of *O. niloticus*.** A total of 816,569 high-quality reads were obtained after DADA2 filtration. The reads were classified into 4 phyla, 8 families, and 21 genera. We found 13 shared amplicon sequence variants (ASVs) among the 3 groups and 9 unique ASVs in the CD (*B. haynesii* CD223) and SM (*A. mimigardefordensis* SM421) groups (Fig. 3A). The alpha diversity measurements showed an enhancement of bacterial

**TABLE 1** Marine soil bacterial isolates identified in different groups based on their colony, morphological, physiological, and biochemical characteristics[a]

| Characteristic and test parameter | Result for bacterial group: | | | |
|---|---|---|---|---|
| | 1 | 2 | 3 | 4 |
| Colony characteristics | | | | |
| Size | Medium | Small | Medium | Small |
| Type | Round | Round | Round | Round |
| Color | Whitish | Yellowish | Creamy | Opaque |
| Shape | Convex | Convex | Convex | Convex |
| Morphological characteristics | | | | |
| Shape | Rod | Coccus | Rod | Coccus |
| Physiological characteristics | | | | |
| Motility | + | + | + | − |
| Growth with 6.5% NaCl | + | + | + | + |
| Biochemical characteristics | | | | |
| Gram staining | + | − | − | + |
| Oxidase | − | + | + | + |
| Catalase | + | + | + | − |
| Oxidative-fermentative | F | O | O | F |
| Motility | + | − | + | − |
| Methyl red | + | + | + | − |
| Voges-Proskauer | − | + | − | − |
| Indole | − | − | + | − |
| $H_2S$ production | + | + | − | − |
| Urease | − | + | V | − |
| Nitrate reductase | + | − | + | − |
| $\beta$-Galactosidase | V | − | − | − |
| Hydrolysis of: | | | | |
| Gelatin | + | + | + | + |
| Esculin | + | + | V | + |
| Casein | + | + | V | + |
| Tween 40 | + | + | + | V |
| Acid production from: | | | | |
| Glycerol | + | − | + | + |
| Galactose | V | − | + | + |
| D-Glucose | V | − | + | + |
| D-Fructose | + | + | V | + |
| D-Mannose | + | − | − | + |
| Mannitol | + | − | V | − |
| Amygdalin | V | − | + | + |
| Maltose | + | − | V | + |

[a]O, oxidative; F, fermentative; +, positive; −, negative; V, variable. Isolates CD223 and SM421 were picked from group 1 and group 3, respectively. The numbers of isolates in the different groups were as follows: 9 in group 1, 8 in group 2, 14 in group 3, and 5 in group 4.

diversity with the SM diet in terms of richness (observed, $P = 0.002$) and the presence of rare species (Chao1, $P = 0.002$). While the Shannon evenness was found to be significantly higher in the SM group than in the CD group and the control group, the Simpson diversity increased in both treatment groups (CD and SM) compared with the control group (Fig. 3B). The beta ordination in terms of unweighted and weighted UniFrac distance metrics showed distinct clustering of bacteria for both treatments compared to the control; however, the dispersion based on presence/absence (unweighted, $P < 0.001$) was found to be more significant than the relative abundance (weighted, $P < 0.002$) (Fig. 3C and D). Overall, the bacterial community was dominated by *Fusobacterium* (92.6%) at the phylum level and *Cetobacterium* (86.8%) at the genus level. SM treatment increased the abundances of *Shewanella* ($P = 0.004$) and *Bacteroides* ($P = 0.051$) and reduced that of *Enterobacter*, which had a higher abundance in the control group ($P = 0.02$) than in the two treatment groups (Fig. 3E and F).

**TABLE 2** Inhibitory activity of marine soil isolates against fish and shrimp pathogens[a]

| Soil isolate | Result with test pathogen | | | | | | |
|---|---|---|---|---|---|---|---|
| | RhG4 | FF11 | F1B1 | B55 | B19 | TS122 | 43BS11 |
| CD223 | − | + | + | − | − | − | − |
| SM421 | − | + | + | − | − | − | − |

[a]−, negative; +, positive; RhG4, *Stenotrophomonas maltophilia* strain; F1B1 and FF11, *Enterococcus faecalis* strains; B55 and B19, *Aeromonas veronii* strains; TS122, *Providencia rettgeri* strain; 43BS11, *Serratia marcescens* strain.

***B. haynesii* CD223 and *A. mimigardefordensis* SM421 prevented *E. faecalis* infection in artificially challenged *O. niloticus*.** To assess the effects of the dietary inclusion of *B. haynesii* CD223 and *A. mimigardefordensis* SM421 on disease prevention in treated *O. niloticus* fish, bacterium-treated fish were artificially infected with *E. faecalis* strains F1B1 and FF11 by an immersion assay. Surprisingly, both groups of marine bacterium-fed fish survived and showed no disease symptoms when challenged with *E. faecalis* F1B1. The rates of survival of the *B. haynesii* CD223- and *A. mimigardefordensis* SM421-fed fish were 80% and 100%, respectively, when artificially infected with *E. faecalis* FF11. However, all of the untreated control fish died, with prominent symptoms of streptococcosis.

**MIC of the ECPs of CD223 and SM421.** The MIC of the ECPs of *B. haynesii* CD223 and *A. mimigardefordensis* SM421 was determined against highly virulent fish-pathogenic strains of *E. faecalis* (strains F1B1 and FF11). The quantitative bioassay revealed that the MIC of the ECPs of strains CD223 and SM421 was 125 $\mu$g mL$^{-1}$ against *E. faecalis* strains F1B1 and FF11 (Table 4).

***In vivo* infection recovery study. (i) *In vivo* curative recovery assay.** To determine the potential applicability of the ECPs of *B. haynesii* CD223 and *A. mimigardefordensis* SM421 as antimicrobial agents, *in vivo* infection recovery studies were carried out. *O. niloticus* fish were immersed in a culture suspension of *E. faecalis* strains F1B1 and FF11. Next, methanolic extracts of the crude bacterial ECPs of both strains were mixed with fish feed and fed separately to *O. niloticus* fish for 7 days at doses of 125, 250, and 500 $\mu$g mL$^{-1}$. For both strains CD223 and SM421, the highest survival rate (100%) was obtained when the fish were challenged with *E. faecalis* strain F1B1 and fed crude ECPs at 250 $\mu$g mL$^{-1}$ (treatment 2 [T2] and T10) (Table 5). For *B. haynesii* CD223, the rates of survival of fish challenged with *E. faecalis* strain F1B1 and fed crude ECPs at 125 $\mu$g mL$^{-1}$ (T1) and 500 $\mu$g mL$^{-1}$ (T3) were 60% and 80%, respectively; for *A. mimigardefordensis* SM421, the values were 70% (T9) and 90% (T11), respectively (Table 5). Only 30% of the control group fish (T4 and T12) survived when challenged with

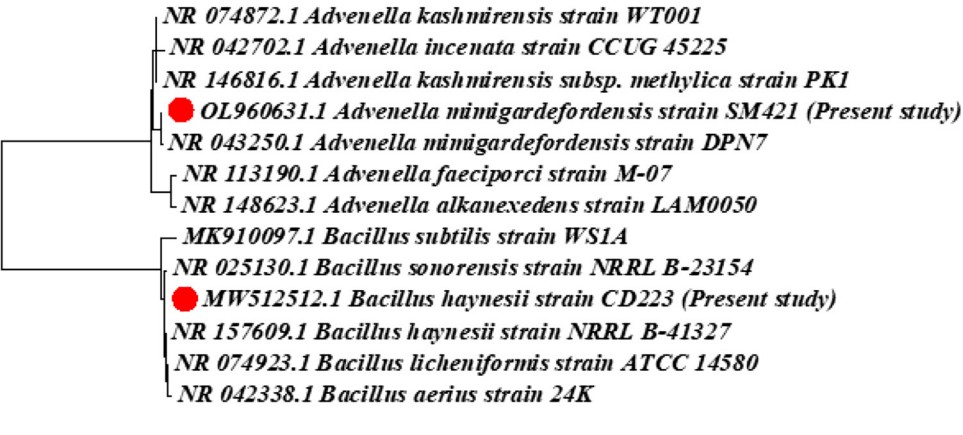

**FIG 1** Phylogenetic positions of the marine soil isolates *B. haynesii* CD223 and *A. mimigardefordensis* SM421. The 16S rRNA gene sequence-based phylogenetic reconstruction was based on the neighbor-joining method with 1,000 bootstrap replicates. The red solid circles indicate the strains used in this study.

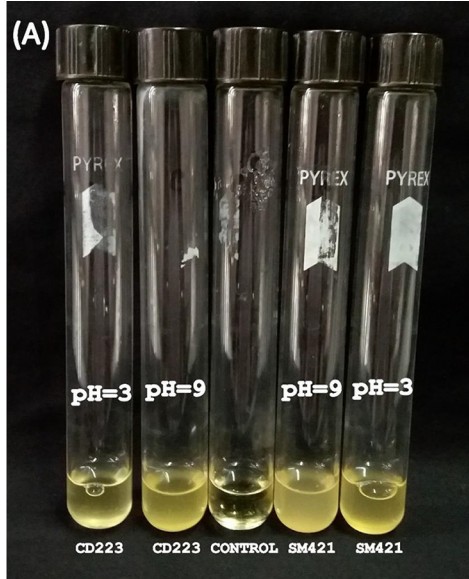

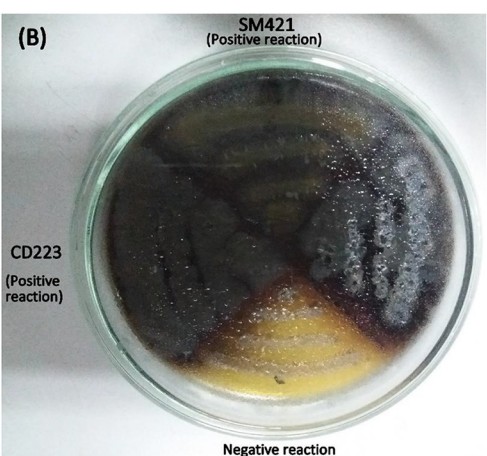

**FIG 2** Assessment of the viability of *B. haynesii* strain CD223 and *A. mimigardefordensis* strain SM421 at different pH levels (range, 3 to 9) and their bile-hydrolyzing activity. (A) pH tolerance test. (B) Bile-hydrolyzing activity test.

*E. faecalis* strain F1B1. In contrast, when the fish were challenged with *E. faecalis* strain FF11, the highest survival rate (90%) was obtained in fish fed crude ECPs at a rate of 250 $\mu$g mL$^{-1}$ (T6 and T14), and only 20% of the control group fish (T8 and T16) survived when challenged with *E. faecalis* strain FF11. No disease symptoms were observed in the treated fish, whereas distinct disease symptoms were observed in the challenged control group fish (Fig. 4). The symptoms of streptococcosis observed were opaque and protruded eyes, a slimy body, a swollen abdomen with tail and fin erosion, and hemorrhaging at the mouth, opercula, fins, and anus. The significant beneficial effects of the ECPs of strains CD223 and SM421 were also supported by nested analysis of variance (ANOVA) (Table 6).

**(ii) *In vivo* prophylactic recovery assay.** To identify the prophylactic potential of *B. haynesii* CD223 and *A. mimigardefordensis* SM421, *in vivo* assays were performed. Methanolic extracts of crude bacterial ECPs were mixed with commercial pelleted feed and fed to *O. niloticus* fish for 7 days at a dose of 250 $\mu$g mL$^{-1}$. After 7 days, *O. niloticus* fish were immersed in a culture suspension of *E. faecalis* strains F1B1 and FF11. The bacterial methanolic extract of the ECP-mixed feed was fed to the fish for another 7 days. In this study, the highest survival rate (100%) was obtained for both *B. haynesii* CD223 and *A. mimigardefordensis* SM421 when the fish were artificially challenged with *E. faecalis* strain F1B1 and fed crude ECPs at 250 $\mu$g mL$^{-1}$. On the other hand, a survival rate of 90% and 100% were obtained in the fish challenged with *E. faecalis* strain FF11 and fed crude ECPs of *B. haynesii* CD223 and *A. mimigardefordensis* SM421, respectively at a rate of 250 $\mu$g mL$^{-1}$ (Table 7). However, only 20% of the

**TABLE 3** Effects of *B. haynesii* strain CD223 and *A. mimigardefordensis* strain SM421 on the growth performance and hematological parameters of *O. niloticus*[a]

| Parameter | Mean value ± SD for group | | |
| --- | --- | --- | --- |
| | Control fish | CD223-fed fish | SM421-fed fish |
| BWG (%) | 48.28 ± 2.21 C | 114.59 ± 3.84 B | 221.68 ± 9.76 A |
| SGR (%/day) | 0 0.62 ± 0.12 C | 1.23 ± 0.13 B | 1.92 ± 0.11 A |
| Hemoglobin concn (g dL$^{-1}$) | 4.33 ± 0.4 B | 6.17 ± 0.1 A | 3.47 ± 0.4 C |
| WBC count (10$^7$ mL$^{-1}$) | 2.29 ± 0.2 C | 3.65 ± 0.2 A | 2.38 ± 0.7 B |
| RBC count (10$^5$ mL$^{-1}$) | 0.78 ± 0.2 C | 1.00 ± 0.2 B | 2.83 ± 0.09 A |
| IgM concn ($\mu$g dL$^{-1}$) | 70.00 ± 4.6 C | 101.67 ± 5.3 B | 108.33 ± 3.3 A |

[a]Values are expressed as means ± standard deviations. Different letters in the rows indicate significant differences by an LSD test ($P < 0.05$) ($n = 3$). BWG, body weight gain; SGR, specific growth rate; WBC, white blood cell; RBC, red blood cell; IgM, immunoglobulin M.

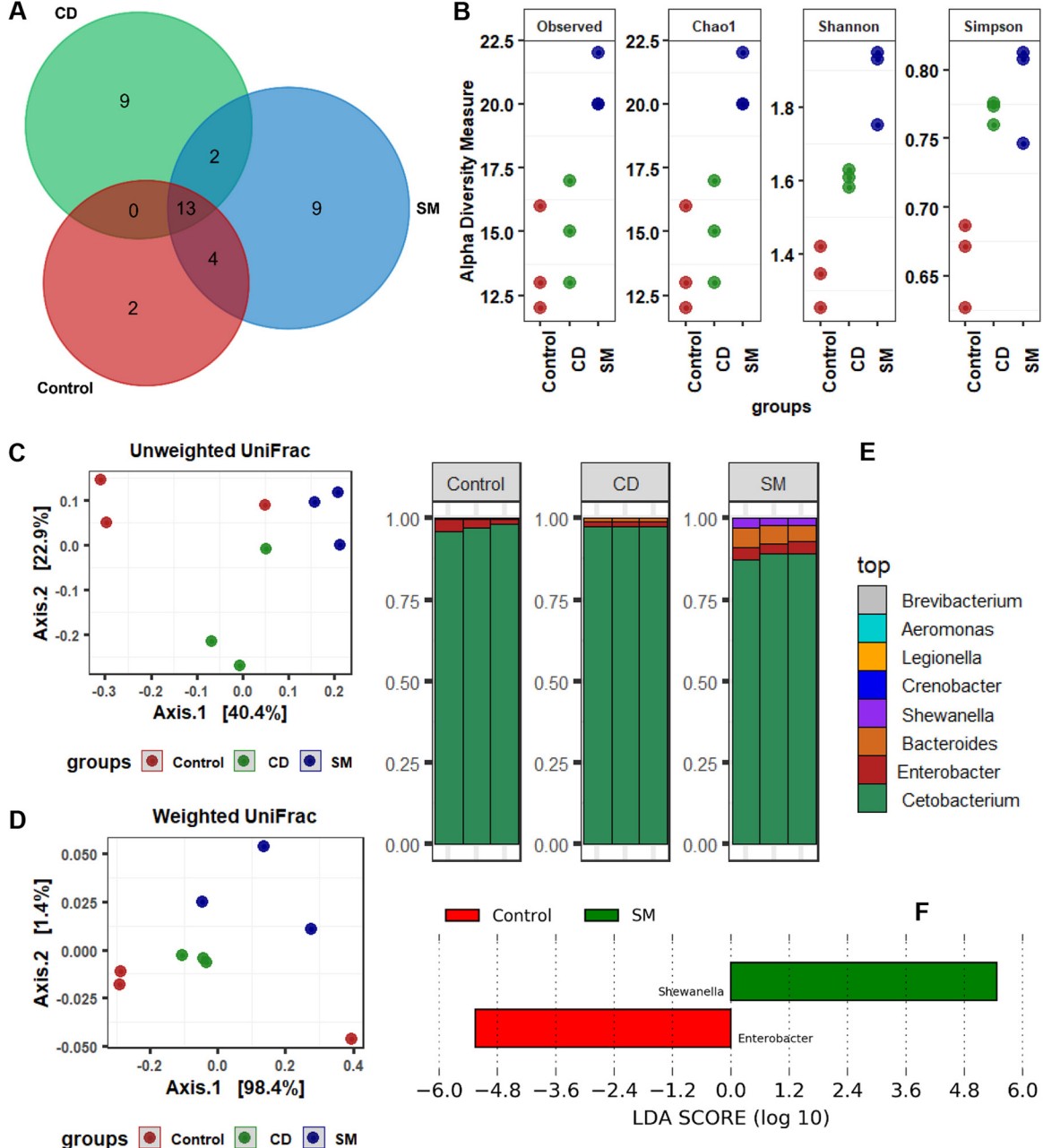

**FIG 3** Gut microbial communities in tilapia fish after feeding with ECPs of *B. haynesii* strain CD223 and *A. mimigardefordensis* strain SM421. (A) Numbers of shared and unique classified ASVs in three different groups. (B) Alpha diversity measurements in terms of richness (observed species and Chao1) and evenness (Shannon and Simpson). (C and D) Beta ordination based on unweighted (presence/absence) (C) and weighted (abundance) (D) UniFrac distance metrics. (E) Relative abundances of bacteria at the genus level in the three different groups. (F) Significantly different genera in the fish gut with LDA values of ≥2.0 and *P* values of <0.05. Abbreviations: CD, *Bacillus haynesii* strain CD223; SM, *Advenella mimigardefordensis* strain SM421.

control group fish challenged with *E. faecalis* survived in both treatment groups. No disease symptoms were observed in the treated *O. niloticus* fish.

## DISCUSSION

The inappropriate application and overuse of different antibiotics in aquaculture have propelled the rapid emergence of numerous multidrug-resistant (MDR) fish pathogens globally. Accordingly, fish diseases are becoming the major limiting factor for the development of aquaculture worldwide. Therefore, there is an urgent need for new antibiotic-producing beneficial bacterial strains that can effectively prevent and control

**TABLE 4** MICs of the ECPs of *B. haynesii* strain CD223 and *A. mimigardefordensis* strain SM421 against fish-pathogenic *E. faecalis* strains[a]

| *E. faecalis* strain and ECP concn ($\mu$g/mL) | Inhibition zone ratio of *B. haynesii* strain CD223 | Inhibition zone ratio of *A. mimigardefordensis* strain SM421 |
|---|---|---|
| F1B1 | | |
| 1,000 | 2.5 | 2.8 |
| 500 | 2 | 2.2 |
| 250 | 1.5 | 1.8 |
| 125 | 1.2 | 1.3 |
| 62.5 | | |
| 31.25 | | |
| | | |
| FF11 | | |
| 1,000 | 2.5 | 2.7 |
| 500 | 2 | 2.2 |
| 250 | 1.5 | 1.8 |
| 125 | 1.2 | 1.2 |
| 62.5 | | |
| 31.25 | | |

[a]The MICs of the ECPs against *E. faecalis* strains F1B1 and FF11 were determined to be 125 $\mu$g/mL.

emerging MDR fish pathogens. Marine bacteria have the potential to produce a wide array of novel bioactive natural products, such as antibacterial, antiviral, antifungal, antiparasitic, antitumor, and antioxidant compounds (37). In the present study, we discovered two bacteria, *viz.*, *B. haynesii* strain CD223 and *A. mimigardefordensis* strain SM421, from the sediments of Saint Martin's Island in the Bay of Bengal that significantly inhibited MDR isolates of a destructive fish pathogen, *E. faecalis*. The application of both live bacteria and their ECPs has been shown to protect fish from streptococcosis when the fish were previously artificially challenged with highly virulent strains of *E. faecalis*. The suppressive effects of some marine bacteria isolated from marine sponges from Saint Martin's Island on fish disease have recently been reported. The present study demonstrated, for the first time, that two soil-derived marine bacteria from the same island suppressed the MDR fish-pathogenic bacterium *E. faecalis* and prevented streptococcosis in Nile tilapia.

**TABLE 5** Curative effect of the ECPs of *B. haynesii* strain CD223 and *A. mimigardefordensis* strain SM421 on *O. niloticus* against streptococcosis caused by *E. faecalis*

| Marine bacterium against pathogenic strain | Treatment | Pathogen concn (CFU/mL) | ECP concn ($\mu$g/mL) | Rate of survival (%) |
|---|---|---|---|---|
| *B. haynesii* strain CD223 against: | | | | |
| *E. faecalis* strain F1B1 | T1 | $1.8 \times 10^7$ | 125 | 60 |
| | T2 | $1.8 \times 10^7$ | 250 | 100 |
| | T3 | $1.8 \times 10^7$ | 500 | 80 |
| | T4 (control) | $1.8 \times 10^7$ | | 30 |
| *E. faecalis* strain FF11 | T5 | $2.3 \times 10^7$ | 125 | 60 |
| | T6 | $2.3 \times 10^7$ | 250 | 90 |
| | T7 | $2.3 \times 10^7$ | 500 | 80 |
| | T8 (control) | $2.3 \times 10^7$ | | 20 |
| | | | | |
| *A. mimigardefordensis* strain SM421 against: | | | | |
| *E. faecalis* strain F1B1 | T9 | $1.8 \times 10^7$ | 125 | 70 |
| | T10 | $1.8 \times 10^7$ | 250 | 100 |
| | T11 | $1.8 \times 10^7$ | 500 | 90 |
| | T12 (control) | $1.8 \times 10^7$ | | 30 |
| *E. faecalis* strain FF11 | T13 | $2.3 \times 10^7$ | 125 | 60 |
| | T14 | $2.3 \times 10^7$ | 250 | 90 |
| | T15 | $2.3 \times 10^7$ | 500 | 80 |
| | T16 (control) | $2.3 \times 10^7$ | | 20 |

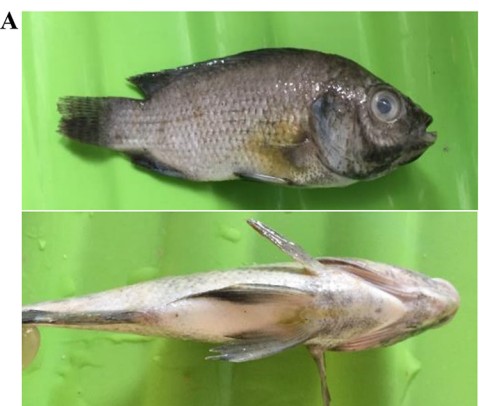 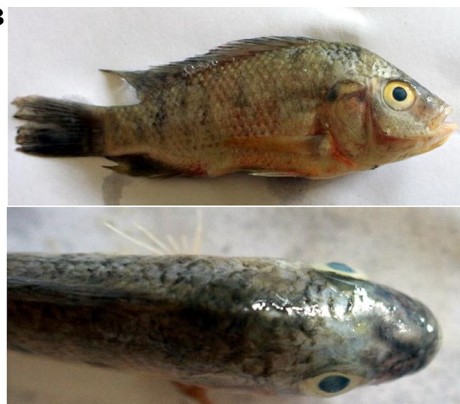

**FIG 4** *In vivo* infection challenge test with highly virulent *E. faecalis*. (A) Fish fed ECPs of *B. haynesii* strain CD223 and infected with *E. faecalis* strain FF11 with no disease symptoms. (B) Control group fish artificially infected with *E. faecalis* strain FF11 exhibiting streptococcosis.

*E. faecalis* strains FF11 and F1B1 were previously identified from diseased *O. niloticus* fish in Bangladesh and were shown to have developed antibiotic resistance to amoxicillin, ampicillin, cefuroxime, erythromycin, and penicillin G (10). In the present study, the marine soil bacteria *B. haynesii* CD223 and *A. mimigardefordensis* SM421 exhibited inhibitory activities against these strains both *in vitro* and *in vivo*. The marine soil isolates CD223, and SM421 were shown to belong to the genera *Bacillus* and *Advenella*, respectively, as confirmed by several biochemical tests and 16S rRNA gene sequence analyses. Different phenotypic characteristics, such as the bile esculin-hydrolyzing activity, antimicrobial activity, noncytotoxicity, enhancement of nonspecific immunity (IgM), and promotion of other hematological parameters, revealed that *B. haynesii* CD223 and *A. mimigardefordensis* SM421 are potentially beneficial marine bacterial strains for controlling disease occurrence in aquaculture. The application of the ECPs of *B. haynesii* CD223 and *A. mimigardefordensis* SM421 at a dose of 250 $\mu$g mL$^{-1}$ in fish feed for 7 days significantly prevented the occurrence of streptococcosis in *O. niloticus*, with the ECPs acting as both therapeutic and preventive agents. The prevention of other fish diseases using beneficial bacteria has been studied by many researchers (35, 38–40). However, until now, there have been no reports on the prevention of streptococcosis caused by *E. faecalis* via the application of beneficial marine bacteria.

*B. haynesii* is a species of *Firmicutes* that was first isolated from soil in Evolution Canyon III in Israel (41). *A. mimigardefordensis* (formerly *Tetrathiobacter mimigardefordensis*) is a betaproteobacterium that was first isolated from compost and exhibits the remarkable capacity to utilize organosulfur compounds as the sole carbon source for growth (42, 43). To the best of our knowledge, these bacteria have never been investigated for their role in fish disease prevention and growth promotion. Further study is warranted to elucidate the molecular mechanism underlying their disease prevention efficacy in host organisms.

The dietary inclusion of probiotics and other beneficial microbes positively influences the growth, digestive enzyme activation, host immune response, and disease resistance of *O. niloticus* (44–47). Among beneficial bacteria, *Bacillus* species are widely

**TABLE 6** Nested ANOVA based on the response variable survival and three factors, marine bacteria, pathogens, and ECPs[a]

| Factor(s) | df | Sum of squares | Mean square | F value | $P_r$(>F) |
|---|---|---|---|---|---|
| Marine bacteria | 1 | 208 | 208.3 | 2.041 | 0.1628 |
| Marine bacteria, pathogens | 2 | 750 | 375 | 3.673 | 0.0366 |
| Marine bacteria, ECPs | 6 | 33,183 | 5,531 | 52.672 | 0.0000 |
| Marine bacteria, pathogens, ECPs | 12 | 33,367 | 2,780.6 | 27.238 | 0.0000 |

[a]df, degree of freedom. The *P* value for marine bacteria plus ECPs was lower than that for marine bacteria plus pathogens and ECPs.

**TABLE 7** Preventive effect of the ECPs of *B. haynesii* strain CD223 and *A. mimigardefordensis* strain SM421 against streptococcosis caused by *E. faecalis*

| *E. faecalis* strain and treatment | Type of bacterial isolate | Bacterial concn (CFU/mL) | ECP concn ($\mu$g/mL) | Rate of survival (%) |
|---|---|---|---|---|
| F1B1 | | | | |
| Positive control | | $1.5 \times 10^7$ | | 20 |
| Negative control | | | | 100 |
| ECPs of CD223 | Normal | $1.5 \times 10^7$ | 250 | 100 |
| ECPs of SM421 | Normal | $1.5 \times 10^7$ | 250 | 100 |
| | | | | |
| FF11 | | | | |
| Positive control | | $2.3 \times 10^7$ | | 20 |
| Negative control | | | | 100 |
| ECPs of CD223 | Normal | $2.3 \times 10^7$ | 250 | 90 |
| ECPs of SM421 | Normal | $2.3 \times 10^7$ | 250 | 100 |

used for their beneficial effects on immunity and health and have been reported to be promising probiotics for Nile tilapia (*O. niloticus*) (45, 46, 48). Members of the genus *Bacillus* are well known for their ability to produce numerous antimicrobial peptides such as the polyketides difficidin, subtilin, subtilosin A, bacillibactin, macrolactin, surfactin, bacilysin, bacillaene, and fengycin, which can protect host organisms through the induction of systemic resistance by inducing the production of stress-related metabolites and the expression of stress response genes (49–51). The potential effects of beneficial *Bacillus* spp. on growth enhancement, improvement of host immunity, and disease prevention have been extensively investigated in many aquacultured species (52). This is the first report on the assessment of the beneficial role of *A. mimigardefordensis* in growth promotion and disease prevention in fish. However, this species is well known for its ability to catabolize the xenobiotic polythioester (53).

In the present study, the application of *B. haynesii* CD223 and *A. mimigardefordensis* SM421 as feed supplements ($1 \times 10^6$ CFU mL$^{-1}$) for *O. niloticus* significantly ($P < 0.05$) enhanced morphometric characteristics such as weight gain (percent) and the SGR (percent per day) compared to the control (Table 3). These findings suggested that *B. haynesii* CD223 and *A. mimigardefordensis* SM421 enhanced growth and caused physiological changes in *O. niloticus*, similar to the findings of other researchers (54, 55). These positive changes may be attributed to the increase in digestive enzyme activity as well as the enhancement of metabolic responses (56, 57). In addition, *B. haynesii* CD223 and *A. mimigardefordensis* SM421 enhanced the RBC, WBC, IgM, and hemoglobin levels in the blood of the treated *O. niloticus* fish. Therefore, the treated *O. niloticus* fish might have been able to consume a sufficient amount of dissolved oxygen (DO) for daily respiration. Moreover, the enhanced WBC and IgM levels in the blood also enhanced the immunity of *O. niloticus* via macrophages and neutrophils, which might help prevent *E. faecalis* infection. The ECPs of *B. haynesii* CD223 and *A. mimigardefordensis* SM421 may also have increased the immunoglobulin level by stimulating the proliferation of T and B lymphocytes in *O. niloticus*, which is consistent with the findings of Dohail et al. (58) and Picchietti et al. (59).

Gut microbiota analysis has become a robust and popular tool to assess the effects of dietary supplementation on aquaculture. In this study, *O. niloticus* fish were cultured in a controlled environment (water temperature, pH, DO level, and ammonia concentration). Hence, gut microbial differences were observed in terms of the antibacterial activity of the ECPs of the beneficial bacteria *B. haynesii* CD223 and *A. mimigardefordensis* SM421. Investigation of the gut microbiota in all of the treatment groups clearly showed that *B. haynesii* CD223 and *A. mimigardefordensis* SM421 modulated the intestinal microbial communities by reducing the load of pathogenic *Enterococcus* spp. in the gut, which ultimately induced the formation of healthy gut microflora. These bacterial communities might have the ability to synthesize ECPs that activate digestive enzymes to promote digestion in the host (60). Moreover, the overwhelming

abundance of *Cetobacterium* in the gastrointestinal tract of *O. niloticus* demonstrated that it constitutes the core microbiota of *O. niloticus* and plays an impactful role in immunity and digestion. In recent decades, several studies have also documented that anaerobic bacterial communities of *Cetobacterium* are commonly found in the gastrointestinal tract of many fish species and are considered mostly beneficial for their role in microbe-host interactions and digestion (61–64). The oral administration of the ECPs of *A. mimigardefordensis* SM421 significantly increased the abundance of the genus *Shewanella*, members of which are also considered beneficial bacteria due to their role in growth enhancement and disease prevention in fish (65).

In summary, the application of two newly identified marine bacteria, *B. haynesii* CD223 and *A. mimigardefordensis* SM421, and their ECPs through feed significantly improved growth, the immune response, and the gut microbiota composition and led to the development of disease resistance against streptococcosis in *O. niloticus*, exhibiting both prophylactic and therapeutic effects. Taken together, our results suggest that *B. haynesii* strain CD223 and *A. mimigardefordensis* strain SM421 are two marine bacteria with potential applications in the biocontrol of streptococcosis in aquacultured Nile tilapia. A further large-scale study is needed to evaluate the efficacy of these bioagents and elucidate their mode of action in disease control in Nile tilapia.

## MATERIALS AND METHODS

**Ethical statement.** The number of animals used was kept to the absolute minimum required to achieve statistical significance for validation purposes. All procedures were conducted in accordance with the United Kingdom Animal (Scientific Procedures) Act of 1986 and ARRIVE 2.0 guidelines (66) and were approved by the Ethical Review Committee (ERC) of the Institute of Biotechnology and Genetic Engineering (IBGE), Bangabandhu Sheikh Mujibur Rahman Agricultural University (BSMRAU), Gazipur, Bangladesh (reference number BSMRAU/IBGE/ERC/001).

**Euthanasia methods.** Experimental fingerlings of Nile tilapia (*O. niloticus*) were euthanized using pure clove oil (catalog number C8392; Sigma) as described previously by Fernandes et al. (67). Briefly, an emulsion of clove oil was prepared by mixing it with ethyl alcohol in a 1:9 ratio (clove oil-ethyl alcohol). Next, the emulsion was diluted in water to obtain concentrations of 0.05 mL (50 mg) and 0.20 mL (200 mg) of clove oil per 500 mL of water. For blood collection, experimental fish were anesthetized by using 0.05 mL of clove oil per 500 mL of water. For the metagenomics study, fish were humanely killed by using 0.20 mL of clove oil per 500 mL of water, and finally, death was ensured by the destruction of the brain (67, 68).

**Collection of experimental fingerlings.** Experimental fingerlings of *O. niloticus* (average weight of 30 ± 1.2 g) were acquired from a fish farm in Mymensingh, Bangladesh, with no previous report of disease occurrence and transported under live conditions to the wet laboratory of IBGE, BSMRAU, under sufficient oxygenation. The handling and housing of all the experimental fingerlings were performed according to guidelines for the care and use of fish by the Canadian Council on Animal Care (68). Water quality parameters (temperature, pH, and DO level) were routinely measured during the acclimatization period (15 days) to maintain the good health of the fingerlings. The fingerlings were fed commercial pelleted fish feed (Mega Feed Company Ltd., Bangladesh) containing 13% moisture, 31% protein, 7% lipid, 13% ash, and 11% fiber at 5% of their body weight until further use.

**Collection and pretreatment of soil samples.** During low tide, marine soil samples were collected from the shore of Saint Martin's Island (20°37′40.3″N, 92°19′22.3″E) in the Bay of Bengal, Bangladesh. To avoid collecting undesirable microbes present on the soil surface, the samples were collected from a depth of 6 to 15 in. using an auger and immediately transferred to a 50-mL sterile Falcon tube. Next, the collected samples were transported to the Advanced Molecular Biology Laboratory of the IBGE, BSMRAU, for the isolation of bacteria. Prior to inoculation, the soil samples were aseptically dried at 120°C for 60 min to eliminate undesirable bacteria (69). Next, the dried soils were ground finely using a sterile mortar and pestle.

**Isolation, phenotypic identification, and maintenance of bacteria.** Bacteria were isolated by serial dilution (up to $10^{-3}$) according to methods described previously by Mohseni et al. (70). Briefly, 1 g of each soil sample was suspended in 9 mL of autoclaved seawater in an individual test tube by using a vortex mixer (Stuart SA7; Bibby Scientific Ltd., Staffordshire, UK). Next, 100-$\mu$L suspensions from each diluted stock ($10^{-2}$ and $10^{-3}$) were aseptically inoculated onto individual starch casein agar (SCA) (HiMedia Laboratories Pvt. Ltd., India) plates in a laminar airflow cabinet (Esco Micro Pte. Ltd., Singapore). The SCA medium was supplemented with cycloheximide (100 mg $L^{-1}$) and nalidixic acid (20 mg $L^{-1}$) to minimize fungal contamination (71). The inoculated plates were incubated at 27°C ± 2°C for 7 days and observed intermittently during incubation. After incubation, different colonies were picked based on their characteristics (color, shape, and size) and purified by subculturing them onto the same medium by the streak culture method. The shapes of the bacterial isolates were observed under a light microscope (Primo Star; Carl Zeiss, Germany) after Gram staining (72). Biochemical tests, such as oxidase, catalase, motility, and oxidative-fermentative (O-F) tests, were accomplished according to methods described previously (35, 73) for the phenotypic characterization of the marine bacterial isolates.

The bacterial stocks were maintained in nutrient broth (NB) supplemented with 10% glycerol and stored in a freezer (Thermo Fisher Scientific, USA) at $-80°C$.

**Test fish pathogens.** Seven MDR high-virulence fish and shrimp pathogens belonging to five genera were used as test organisms for the *in vitro* screening of the antimicrobial activity of soil bacteria. These strains were *Stenotrophomonas maltophilia* strain RhG4 (74), *E. faecalis* strains F1B1 and FF11 (10), *Aeromonas veronii* strains B55 and B19 (75), *Providencia rettgeri* strain TS122, and *Serratia marcescens* strain 43BS11 (76). All of the strains were isolated from diseased fish and shrimp from Bangladesh and stored at the laboratory of the IBGE, BSMRAU.

**Screening for antagonistic activity of marine soil isolates.** To assess the antimicrobial activity of soil isolates, bacteria were grown in marine broth (MB) 2216 (Merck, USA) for 7 days at 28°C. The broth culture was centrifuged at $10,000 \times g$ for 15 min, and the culture supernatant was passed through a 0.22-$\mu$m Millipore membrane filter (Sigma-Aldrich, USA). The inhibitory activity of the culture supernatant was determined by an agar well diffusion assay as described previously by Rakshanya et al. (77). Briefly, 30 $\mu$L of pathogenic isolates ($3 \times 10^5$ CFU mL$^{-1}$) was spread onto Mueller-Hinton agar (Liofilchem, Italy) plates, and the filtered supernatants (50 $\mu$L) of marine soil bacterial isolates were inoculated into a hole made on the agar plates. The plates were incubated at 28°C for 24 h. At the end of the incubation period, the presence of an inhibition zone surrounding the agar wells was recorded.

**Molecular identification of marine soil isolates.** The genomic DNA of the selected isolates was extracted using a commercial DNA purification kit (Thermo Fisher Scientific, USA) according to the manufacturer's instructions. The extracted DNA was quantified using a NanoDrop spectrophotometer (Thermo Fisher Scientific, USA). PCR for the amplification of the targeted 16S rRNA genes of the bacteria was performed with the universal primer pair 8F (5′-AGAGTTTGATCCTGGCTCAG-3′) and 1492R (5′-GGTTACCTTGTTACGACTT-3′) according to the method described previously by Paul et al. (35). Briefly, an initial denaturation step at 94°C for 5 min was followed by 35 cycles of denaturation at 94°C for 1 min, annealing at 57°C for 40 s, and extension at 72°C for 1 min and then a final extension step at 72°C for 10 min. Next, the amplified PCR products were purified using a PCR purification kit (Thermo Fisher Scientific, USA). Next, 16S rRNA gene sequencing was completed at the National Institute of Biotechnology (NIB), Dhaka, Bangladesh. The sequence received in a raw format was then assembled to generate contigs using MEGA 7.0 (78) and primarily edited using BioEdit (version 7.2) to remove the overlapping sequences and chimeras. 16S rRNA gene sequence homology was studied using the Basic Local Alignment Search Tool (BLAST) program of the National Center for Biotechnology Information (NCBI), and phylogenetic analysis was performed by using the neighbor-joining method in MEGA 7.0 (78).

**Assessment of bile esculin activity and evaluation of the viability of bacteria at different pH values.** The bile esculin activity of the strains was evaluated according to methods described previously by Mathew et al. (79). Briefly, isolates CD223 and SM421 were inoculated individually into NB and incubated at 28°C for 48 h. After incubation, the broths were centrifuged at $5,000 \times g$ for 10 min to discard the culture supernatant. The pellets were washed and resuspended in phosphate-buffered saline (PBS) buffer (pH 7.2). Next, the supernatant was discarded by centrifugation at $5,000 \times g$ for 10 min. The pellets of CD223 and SM421 were then carefully taken and inoculated onto bile esculin agar (BEA) (HiMedia, India) plates and incubated at 37°C for 24 h. At the end of the incubation period, the presence of growth and blackening of the BEA were considered a positive reaction.

The viability of *B. haynesii* CD223 and *A. mimigardefordensis* SM421 was assessed under both acidic and alkaline conditions (pH range of 3 to 9) according to the method described previously by Borah et al. (80), with a few modifications. Briefly, the pH of the SCA broth was adjusted from pH 3 to 9. Next, *B. haynesii* CD223 and *A. mimigardefordensis* SM421 were inoculated into this broth and incubated for 24 h at 28°C. The viability of the cells was then confirmed by the spread plate method by inoculating them onto SCA plates (pH 7).

**Preparation of bacterial ECPs.** Bacterial ECPs were prepared as described previously by Paul et al. (35), with some modifications. Briefly, *B. haynesii* CD223 and *A. mimigardefordensis* SM421 were enriched in MB at 28°C for 10 days on an orbital shaker (WiseShake SHO-1D) at 190 rpm. The ECPs were harvested, mixed with an equal volume of ethyl acetate in a separatory funnel, and shaken for 25 min. Next, the upper phase was collected and evaporated in a rotary evaporator (IKA RV-10; IKA-Werke GmbH & Co. KG, Germany) at 42°C. After that, the concentrated ethyl acetate extract was suspended in 20 mL of a methanol–*n*-hexane (1:1) solution in a separatory funnel and gently shaken for 20 min. Finally, the lower phase was collected and air dried for 2 days in a laminar airflow cabinet (Esco Micro Pte. Ltd., Singapore). Next, the air-dried extract was weighed and dissolved in methanol for further use.

**Brine shrimp lethality assay of ECPs.** The cytotoxic effects of the ECPs of *B. haynesii* CD223 and *A. mimigardefordensis* SM421 on axenic brine shrimp (*Artemia salina*) nauplii were determined as described previously (81). Briefly, brine shrimp cysts were immersed in an autoclaved marine water solution with a constant supply of oxygen for 48 h for hatching. Next, 1,000, 500, 250, 125, 62.5, and 31.25 $\mu$g of the ECP extracts of CD223 and SM421 were individually added to 1 mL (100 cells mL$^{-1}$) of a hatched brine shrimp nauplius (at developmental stage II) solution in 24-well plates and kept on an orbital shaker at 100 rpm for 24 h. Each of the treatments and controls were performed in triplicate. After incubation, the survival of the brine shrimp nauplii was recorded.

**MIC determination.** The MIC of the ECPs of *B. haynesii* CD223 and *A. mimigardefordensis* SM421 was determined by the agar dilution method (35). In this regard, sterilized paper discs (10 mm in size) were immersed in several concentrations of ECPs for 10 min. The discs were then air dried for 24 h for the complete evaporation of the solvents. Next, 50-$\mu$L inocula of fish-pathogenic *E. faecalis* strains F1B1 and FF11 were individually spread onto Mueller-Hinton agar plates by the spread culture method. The discs holding different concentrations of ECP extracts (1,000, 500, 250, 125, 62.5, and 31.25 $\mu$g mL$^{-1}$) of the

bacterial strains were then carefully placed onto the surface of the bacterial culture plates and incubated at 28°C for 24 h. After incubation, the culture plates were observed carefully, and the MICs of the ECPs of CD223 and SM421 were calculated as described previously by Paul et al. (35).

***In vivo* growth performance of *O. niloticus* upon dietary inclusion of *B. haynesii* CD223 and *A. mimigardefordensis* SM421.** To assess the beneficial effects of *B. haynesii* CD223 and *A. mimigardefordensis* SM421 on the growth parameters, nine plastic tanks (300 L) were divided into three treatments, F-T1 (control), F-T2 (*B. haynesii* CD223), and F-T3 (*A. mimigardefordensis* SM421), with three replicates each. A total of 180 uniformly sized (average weight of 30 $\pm$ 1.2 g) premature *O. niloticus* fingerlings were randomly distributed into the three treatments, where the stocking density was maintained at 20 fish per tank. Before feeding with the experimental feed, the initial weight of all of the fish was recorded by a digital weight machine (Vision, Bangladesh). The experimental feed was given to the fish twice a day at 5% of their body weight. The unutilized feed and fecal matter were collected every morning, and the water was changed once a day by siphoning. Underground water was used for the experiment. The tanks were maintained at an ambient temperature of 28°C $\pm$ 1°C at pH 7.3 to 8.2 under constant aeration (DO level, 5 $\pm$ 0.5 mg L$^{-1}$).

Commercial pelleted feed (Mega Feed Company Ltd., Bangladesh) was used as the basal diet for supplementation with *B. haynesii* CD223 and *A. mimigardefordensis* SM421 for 90 days. *B. haynesii* CD223 and *A. mimigardefordensis* SM421 were sprayed aseptically into the feed to provide a final concentration of bacterial cells of approximately 1 $\times$ 10$^6$ CFU mL$^{-1}$. The amount of each bacterial strain in the feed was determined using the spread plate technique as described previously by Zokaeifar et al. (82).

The growth of *O. niloticus* fingerlings was evaluated in terms of weight gain and SGR. Regular sampling was performed at 15-day intervals. Weight gain (percent) and SGR (percent per day) were calculated by using the formulas as described previously by Tekinay and Davies (83): weight gain (%) = [(final weight − initial weight)/initial weight] $\times$ 100 and SGR (%/day) = {[ln(final weight) − ln(initial weight)]/time (day)} $\times$ 100.

**Hematological parameters and nonspecific immunity.** After a 90-day feeding trial, the treated fish were anesthetized with clove oil (0.05 mL per 500 mL of water) for hematological analysis. Blood samples were collected from the fish and taken in a tube containing heparin (which acts as an anticoagulant). The hemoglobin content in the samples was tested by using the EasyTouch CGHb testing kit (Bioptik Technologies Inc., China), wherein collected blood samples were poured into the edge of a strip of the hemoglobin meter before the coagulation of blood, as described previously by Salam et al. (60). The RBCs and WBCs were counted using an improved Neubauer hemocytometer (MarienFeld Company, Germany) under a light microscope (DM 100; Leica, Wetzlar, Germany) according to methods described previously by Hesser (84). A standard RBC dilution pipette and a 1:200 dilution were used to determine the erythrocyte count. Estimation of the immunoglobulin M (IgM) concentration was carried out by using a Humalyzer-3000 analyzer.

**DNA extraction and high-throughput sequencing.** The bacterial DNA from the digesta of the hindgut of the experimental *O. niloticus* fish was extracted using the commercial GeneJET genomic DNA purification kit (Thermo Fisher Scientific, USA) according to the manufacturer's instructions. The extracted DNA was quantified using a NanoDrop 2000cc spectrophotometer (Thermo Fisher Scientific, USA), and the DNA was subsequently diluted with nuclease-free water (Thermo Fisher Scientific, USA) to a final concentration of 30 ng $\mu$L$^{-1}$ for PCR. A total of 50 $\mu$L of PCR mix was prepared by mixing 25 $\mu$L of Hot Start 2$\times$ master mix (New England BioLabs Inc., USA), 2 $\mu$L of template DNA, 1 $\mu$L each of V3 and V4 sequencing primers with Illumina overhang adapters (forward primer 5′-TCGTCGGCAGCGTCA GATGTGTATAAGAGACAGCCTACGGGNGGCWGCAG and reverse primer 5′-GTCTCGTGGGCTCGGAGAT GTGTATAAGAGACAGGACTACHVGGGTATCTAATCC) (part number 15044223 Rev. B), and 21 $\mu$L of nuclease-free water (Thermo Fisher Scientific, USA). Thirty cycles of PCR amplification were performed in a Bio-Rad S100 gradient thermal cycler (Bio-Rad Laboratories Inc., CA, USA). Successful amplification was confirmed by gel electrophoresis in 1% agarose, followed by the cleaning and purification of positive amplicons using AMPure XP beads (Bioline, USA). The amplicons were indexed with a Nextera XT index kit via 12 cycles of secondary PCR according to the standard Illumina protocol (part number 15044223 Rev. B). The samples were sequenced on an Illumina MiSeq platform (Illumina Inc., San Diego, CA, USA) at the Harry Perkins Institute of Medical Research, Western Australia, using a v3 kit (600 cycles, part number MS-102-3003).

**Bioinformatics analyses. (i) Processing of Illumina reads.** The raw sequence data were imported into QIIME2 (v2021.4) for paired-end processing (85). Quality trimming (deionizing) of demultiplexed sequences was performed using the q2-dada2 plug-in (86) with the following parameters: -p-trim-left-l 0, -p-trunc-len-f 280, -p-trim-left-r 0, and -p-trunc-len-r 220. The DADA2 output as a feature frequency of an ASV that represents biological features of the amplicon sequence was taxonomically classified against the SILVA 138 release (87). The chimeric features were removed from the table with the q2-vsearch plug-in using the uchime-denovo tool. The feature ASV table was filtered based on the lowest nonzero frequency of 120. Each sample was set to an even depth of 48,804 for downstream analyses of alpha-beta diversity and microbial composition.

**(ii) Downstream bioinformatics.** The rarefed ASV table, taxonomy table, and metadata were used for alpha-beta diversity analyses with R statistical software (v4.11) (88). The observed species and Chao1 indices were selected to observe the changes in diversity, while the Shannon and Simpson indices were used to calculate evenness among groups. Weighted (relative abundance) and unweighted (presence/absence) UniFrac distance metrics were used to calculate beta diversity. Alpha-beta diversity measurements were performed using the phyloseq (89), microbiomeSeq (https://github.com/umerijaz/microbiomeSeq), microbiome (https://github.com/microbiome/microbiome), and vegan R packages that support the ggplot2

plotting packages (90). The relative abundance of bacteria at various taxonomic levels was calculated with the phyloseq R package.

**Statistical analyses of high-throughput sequence reads.** Analysis of variance (ANOVA) with the Kruskal-Wallis rank test was used to compare alpha diversity values among groups. Centroid analysis of beta dispersion among the groups was performed as permutational multivariate analysis of variance (PERMANOVA) with the vegan R package (91). Bacteria that were significantly differentially abundant at the genus level in the three different groups were identified using the linear discriminant analysis (LDA) effect size (LEfSe) algorithm at an LDA cutoff value of 2.0. At every stage of data analysis, a $P$ value of less than 0.05 was considered to indicate statistical significance.

***In vivo* artificial infection with *E. faecalis* by the immersion method.** At the end of the 90-day feeding trial, 10 fish from each treatment (F-T1, F-T2, and F-T3) were artificially challenged with *E. faecalis* strains F1B1 and FF11 by the immersion method. Pathogen inocula were prepared by adding 1,000 mL of a broth culture into 9,000 mL of sterile water. The number of CFU per milliliter of the inoculum was recorded for each isolate tested. The fingerlings were immersed in a pathogenic suspension for approximately 20 min at room temperature (approximately 28°C) and then placed into an aquarium containing 200 L of freshwater to observe the disease prevention efficacy of each treatment.

***In vivo* therapeutic effects of the ECPs of *B. haynesii* CD223 and *A. mimigardefordensis* SM421. (i) *In vivo* curative recovery assay.** An artificial infection challenge test with *E. faecalis* strains F1B1 and FF11 was carried out using the acclimatized fingerlings of *O. niloticus*. After exposure to the highly virulent *E. faecalis* F1B1 and FF11 suspensions, the *O. niloticus* fingerlings were divided into four treatment groups with three replicates (10 fish per tank) for each strain. To assess the curative effects of *B. haynesii* CD223, the treatments were denoted T1, T2, T3, and T4 (control) for strain F1B1 and T5, T6, T7, and T8 (control) for strain FF11. Similarly, to assess the curative effects of *A. mimigardefordensis* SM421, the treatments were denoted T9, T10, T11, and T12 (control) for strain F1B1 and T13, T14, T15, and T16 (control) for strain FF11. Next, the ECPs of *B. haynesii* CD223 incorporated into pelleted feed were fed to the fingerlings at rates of 125 $\mu$g mL$^{-1}$ (for T1 and T5), 250 $\mu$g mL$^{-1}$ (for T2 and T6), and 500 $\mu$g mL$^{-1}$ (for T3 and T7). Similarly, the ECPs of *A. mimigardefordensis* SM421 incorporated into pelleted feed were fed to the fingerlings at rates of 125 $\mu$g mL$^{-1}$ (for T9 and T13), 250 $\mu$g mL$^{-1}$ (for T10 and T14), and 500 $\mu$g mL$^{-1}$ (for T11 and T15). The control group (T4, T8, T12, and T16) fingerlings were fed commercial pelleted feed with no ECP extracts. Fish were fed at 5% of their body weight for 7 days. Continuous aeration was supplied to the aquarium, and half of the aquarium water was replaced every day during the experimental period.

**(ii) *In vivo* prophylactic assay.** To assess the *in vivo* prophylactic effects of the ECPs of *B. haynesii* CD223 and *A. mimigardefordensis* SM421, the *O. niloticus* fingerlings were divided into three treatments with three replicates (10 fish per tank). *O. niloticus* fingerlings were fed ECP extracts incorporated into commercial pelleted feed at a rate of 250 $\mu$g mL$^{-1}$ for 30 days. After that, the fingerlings were exposed to the pathogenic *E. faecalis* strains F1B1 and FF11 by using the same method as the one described above for the artificial infection study and returned to the aquarium. After exposure to pathogenic isolates, the fish were fed basal diets for 14 days. A control group of fish that was not previously provided any ECP extract was maintained for each strain. During this period, fingerlings were observed for any external disease symptoms and abnormal behaviors. Aeration was maintained throughout the experiment, and approximately 50% of the water from the aquarium was replaced every day during the experiment.

**Statistical analyses.** Typical power analysis for one-way ANOVA was performed using G*Power version 3.0.10 as described previously by Faul et al. (92). Arithmetical data were analyzed using IBM SPSS (v23, 2017). ANOVA and nested ANOVA were used to calculate any significant differences ($P < 0.05$) among variables in the control and treatment groups, and the values were separated by least significance difference (LSD) *post hoc* statistics.

**Data availability.** The 16S rRNA gene sequences of isolates CD223 and SM421 were deposited in the GenBank database with accession numbers MW512512.1 and OL960631.1, respectively.

## ACKNOWLEDGMENTS

We sincerely acknowledge the financial support of the Research Management Wing of Bangabandhu Sheikh Mujibur Rahman Agricultural University, Bangladesh, for conducting the research work under the research project entitled Development of Native Probiotics, Herbal Extracts, and Antimicrobial Agents for Sustainable Management of Major Fish Diseases of Bangladesh.

Md Mahfuzur Rahman contributed to methodology and investigation. Sulav Indra Paul contributed to writing – original draft preparation, formal analysis, data curation, methodology, and investigation. Ashikur Rahman, Md Shameul Haque, and Md Ali Arman Ador contributed to methodology. Md Javed Foysal contributed to formal analysis and data curation and revised the manuscript. Md Tofazzal Islam contributed to writing – reviewing and editing. Md Mahbubur Rahman contributed to conceptualization, methodology, investigation, writing – reviewing and editing, visualization, supervision, funding acquisition, and project administration. All authors gave final approval and agree to be accountable for all aspects of the work.

We declare no potential conflicts of interest with respect to the research, authorship, and/or publication of this article.

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
