## [Reviewer comments · Microbiology Spectrum]

Microbiology Spectrum

Suppression of Streptococcosis and Modulation of the Gut Bacteriome in Nile Tilapia (*Oreochromis niloticus*) by Marine Sediment Bacteria *Bacillus haynesii* and *Advenella mimigardefordensis*

Md. Mahfuzur Rahman, Sulav Indra Paul, Ashikur Rahman, Md. Shameul Haque, Md. Ali Arman Ador, Md. Javed Foysal, Md. Tofazzal Islam, and Md. Mahbubur Rahman

Corresponding Author(s): Md. Mahbubur Rahman, Bangabandhu Sheikh Mujibur Rahman Agricultural University

Review Timeline:

Submission Date:	July 4, 2022
Editorial Decision:	August 17, 2022
Revision Received:	October 29, 2022
Accepted:	November 11, 2022

Editor: Artem Rogovskyy

Reviewer(s): Disclosure of reviewer identity is with reference to reviewer comments included in decision letter(s). The following individuals involved in review of your submission have agreed to reveal their identity: Pabodha Weththasinghe (Reviewer #1); Uriel Rodriguez-Estrada (Reviewer #2)

Transaction Report:

DOI: <https://doi.org/10.1128/spectrum.02542-22>

August 17, 2022

Prof. Md. Mahbubur Rahman
Bangabandhu Sheikh Mujibur Rahman Agricultural University
Institute of Biotechnology and Genetic Engineering
Salna
Gazipur 1706
Bangladesh

Re: Spectrum02542-22 (Suppression of Streptococcosis and Modulation of Gut Bacteriome in Tilapia (*Oreochromis niloticus*) by Marine Sediment Bacteria *Bacillus haynesii* strain CD223 and *Advenella mimigardefordensis* strain SM421)

Dear Prof. Md. Mahbubur Rahman:

Link Not Available

Sincerely,

Artem Rogovskyy

Journals Department
Reviewer comments:

Reviewer #1 (Public repository details (Required)):

I recommend to deposit the raw 16S rRNA gene sequence files and metadata in a public repository.

Reviewer #1 (Comments for the Author):

Manuscript ID: Spectrum02542-22

General comment: The manuscript provides novel information regarding the use of two potential probiotic marine sediment bacterial strains to prevent Streptococcosis in Tilapia. This is a very comprehensive study, and the methodology seems very sound. The study results of significant relevance for the aquaculture industry. However, the manuscript requires some rearrangements in particular to avoid repetitions. The manuscript is also required to be checked for correct scientific English and for grammatical errors. In addition, there are some minor issues (see the comments listed below) that the authors should take into consideration.

Abstract

Comment 1: The study involves various phases (i.e., isolation and characterization of bacteria, determination of cytotoxic effects of the ECPs, in vivo study with tilapia fed isolated bacterial strains, in vivo study with tilapia fed ECPs etc). The authors have directly presented various results without mentioning these phases that they followed. I find this bit confusing. Therefore, I would add them in the abstract.

Introduction

Comment 2: Line 75: Please do not start a sentence with 'But'.

Results

Comment 3: Please avoid repetitions in the 'Results' section. I find many repetitions between 'Results' and 'Materials and methods' sections.

Comment 4: Line 145 and Line 147: I would present these results in a table. You may also add them in Table 3.

Discussion

Comment 5: Lines 242-247: Lengthy sentence.

Comment 6: Lines 286-288: Same information is available in Line 145 and 147. Please avoid repetitions.

Comment 7: Lines 292-294 and Line 328: Did authors measure digestive enzyme activities in Tilapia? I do not see any results regarding this in the manuscript. Even though the isolated strains showed enzymatic activities, we can not say they improved enzyme activities in the intestine of tilapia without measuring digestive enzyme activities in tilapia.

Materials and methods

Comment 8: Line 519: Did the authors extract DNA from the digesta of hindgut? Please mention.

Comment 9: Line 560 and Line 608: Same sub-topic. Please use more specific sub-topics.

Comment 10: Line 583: The authors have used the same abbreviations (i.e., T1, T2 and T3) to denote three other diets in Line 482.

Tables

Comment 11: Please provide footnotes for abbreviations used in the tables.

Comment 12: Table 5: Would it not be better to use a nested ANOVA to analyze the results in this table? If you considered this but decided not to, what were your reasons?

Reviewer #2 (Comments for the Author):

Gram-positive *Enterococcus faecalis* (Ef) belongs to the lactic acid bacteria (LAB) group. Ef is formerly classified as part of the group D streptococcus system. These bacteria is found in a diverse number of environments such as the gastrointestinal tract of healthy humans. However, it is a opportunistic pathogen in humans, causing even life-threatening infections, especially in hospital environments, were naturally high levels of antibiotic resistance Ef contribute to pathogenicity. In aquaculture nutrition field, it is well known that Ef, was isolated from the intestines of healthy humans and demonstrated Immune stimulation properties and adjuvant benefits when administered in a lysed or inactivated form.

It is well known that Nile tilapia, is the most important aquacultured species in the tropical regions of the planet. Although, tilapia tolerate a wide gamma of adverse water quality and external stressors better than most commercial aquaculture species, it is evident that this species has certain susceptibility to specific pathogens. These pathogens produce a diverse number of diseases. Such as Streptococcosis, caused by *Streptococcus iniae* and *S. galactidae* or Septicemia caused by *Aeromonas hydrophila* or *Trichodina*, produced by a protozoan parasite or *Columnaris* caused by the myxobacteria, *Flexibacter columnaris*.

Although this manuscript presents results derived from a high quality research (novelty) and well designed study, there are some relevant concerns related to experimental design. Among these concerns, surge the question of why authors did select *Enterococcus faecalis* as pathogen for this study? If there are other more recurrent pathogens affecting Nile tilapia?. To this respect, reviewer strongly suggest that authors should include (in introduction), a brief paragraph talking about how *E. faecalis* infect Nile tilapia cultures in Bangladesh. I mean, how this pathogen agent is able to reach rearing water of farms, since *E. faecalis* is per sé originated in human intestines hence present in urban waste waters.

The other big concern is related to probiotic microorganisms used in present study. Although *Bacillus haynesii* strain CD223 and *Advenella mimigardefordensis* strain SM421 were isolated from marine soil, they were used as probiotic agents in a freshwater species. In reviewer's opinion, probiotic - prebiotic research, should be targeted to candidate microorganisms highly available in similar environments where the host is reared.

On the other hand, it is strongly recommended all along the manuscript, *Enterococcus faecalis* should be mentioned only at the beginning. Subsequently, it can be mentioned as *E. faecalis*.

Finally, in the manuscript, M&M and discussion are misplaced. Results are misleading since in some parts, it seems that are mixed with M&M.

*Note: Please, see attached manuscript file, in order to see more suggestions and comments.

Staff Comments:

Preparing Revision Guidelines

Please return the manuscript within 60 days; if you cannot complete the modification within this time period, please contact me. If you do not wish to modify the manuscript and prefer to submit it to another journal, please notify me of your decision immediately so that the manuscript may be formally withdrawn from consideration by Microbiology Spectrum.

Manuscript ID: Spectrum02542-22

General comment: The manuscript provides novel information regarding the use of two potential probiotic marine sediment bacterial strains to prevent Streptococcosis in Tilapia. This is a very comprehensive study, and the methodology seems very sound. The study results of significant relevance for the aquaculture industry. However, the manuscript requires some rearrangements in particular to avoid repetitions. The manuscript is also required to be checked for correct scientific English and for grammatical errors. In addition, there are some minor issues (see the comments listed below) that the authors should take into consideration.

Abstract

Comment 1: The study involves various phases (i.e., isolation and characterization of bacteria, determination of cytotoxic effects of the ECPs, in vivo study with tilapia fed isolated bacterial strains, in vivo study with tilapia fed ECPs etc). The authors have directly presented various results without mentioning these phases that they followed. I find this bit confusing. Therefore, I would add them in the abstract.

Introduction

Comment 2: Line 75: Please do not start a sentence with 'But'.

Results

Comment 3: Please avoid repetitions in the 'Results' section. I find many repetitions between 'Results' and 'Materials and methods' sections.

Comment 4: Line 145 and Line 147: I would present these results in a table. You may also add them in Table 3.

Discussion

Comment 5: Lines 242-247: Lengthy sentence.

Comment 6: Lines 286-288: Same information is available in Line 145 and 147. Please avoid repetitions.

Comment 7: Lines 292-294 and Line 328: Did authors measure digestive enzyme activities in Tilapia? I do not see any results regarding this in the manuscript. Even though the isolated strains showed enzymatic activities, we can not say they improved enzyme activities in the intestine of tilapia without measuring digestive enzyme activities in tilapia.

Materials and methods

Comment 8: Line 519: Did the authors extract DNA from the digesta of hindgut? Please mention.

Comment 9: Line 560 and Line 608: Same sub-topic. Please use more specific sub-topics.

Comment 10: Line 583: The authors have used the same abbreviations (i.e., T1, T2 and T3) to denote three other diets in Line 482.

Tables

Comment 11: Please provide footnotes for abbreviations used in the tables.

Comment 12: Table 5: Would it not be better to use a nested ANOVA to analyze the results in this table? If you considered this but decided not to, what were your reasons?

Response to Reviewer

Dear Reviewer,

Thank you so much for your review and specific comments and valuable suggestions to improve the manuscript. We have revised the manuscript according to your valuable comments and suggestions. Our point-by-point responses are listed below.

Reviewer #1

Reviewer's comment:

Reviewer #1 (Public repository details (Required)):

I recommend to deposit the raw 16S rRNA gene sequence files and metadata in a public repository.

Our response:

Thank you very much for the comment. The *16S rRNA* gene sequences of these isolates CD223 and SM421 were deposited in the NCBI data bank with accession numbers MW512512.1, and OL960631.1, respectively. We provided the hyperlink of *16S rRNA* gene sequences in the revised manuscript.

Reviewer's comment:

General comment: The manuscript provides novel information regarding the use of two potential probiotic marine sediment bacterial strains to prevent Streptococcosis in Tilapia. This is a very comprehensive study, and the methodology seems very sound. The study results of significant relevance for the aquaculture industry. However, the manuscript requires some rearrangements in particular to avoid repetitions. The manuscript is also required to be checked for correct scientific English and for grammatical errors. In addition, there are some minor issues (see the comments listed below) that the authors should take into consideration.

Our response:

Thank you so much for your encouraging comments and suggestions. We have minimized the repetitions in this revision and rearranged the manuscript according to all reviewers' suggestions. The English has been checked and revised by a native speaker whereas grammar and language are improved through a professional editing service to increase the readability of the MS (Verification link: <https://secure.aje.com/en/certificate/verify>; Verification Code: DFB1-F83A-D791-8FED-CE9B). Additionally, the American English has been followed throughout the manuscript. Hope, you will find our response and revised MS satisfactory.

Reviewer's comment:

Abstract

Comment 1: The study involves various phases (i.e., isolation and characterization of bacteria, determination of cytotoxic effects of the ECPs, in vivo study with tilapia fed isolated bacterial strains, in vivo study with tilapia fed ECPs etc). The authors have directly presented various results without mentioning these phases that they followed. I find this bit confusing. Therefore, I would add them in the abstract.

Our response:

We have revised the abstract based on the flow of methods used to analyze data emphasizing reviewer's suggestions.

Reviewer's comment:

Introduction

Comment 2: Line 75: Please do not start a sentence with 'But'.

Our response:

We revised it.

Reviewer's comment:

Results

Comment 3: Please avoid repetitions in the 'Results' section. I find many repetitions between 'Results' and 'Materials and methods' sections.

Our response:

Thank you so much for the comment. We revised the manuscript thoroughly and minimized the redundancy in the 'Results' and 'Materials and methods' sections.

Reviewer's comment:

Comment 4: Line 145 and Line 147: I would present these results in a table. You may also add them in Table 3.

Our response:

We revised it according to the reviewer's suggestion.

Reviewer's comment:

Discussion

Comment 5: Lines 242-247: Lengthy sentence.

Our response:

Revised.

Reviewer's comment:

Comment 6: Lines 286-288: Same information is available in Line 145 and 147. Please avoid repetitions.

Our response:

Repetitive information has been removed in the revised MS.

Reviewer's comment:

Comment 7: Lines 292-294 and Line 328: Did authors measure digestive enzyme activities in Tilapia? I do not see any results regarding this in the manuscript. Even though the isolated strains showed enzymatic activities, we can not say they improved enzyme activities in the intestine of tilapia without measuring digestive enzyme activities in tilapia.

Our response:

Thank you so much for the valuable comment. We didn't measure the digestive enzyme activities in Tilapia. We only performed the enzymatic activities of the isolated strains. Yes, we agree with your comments. Therefore, in this revision, we removed the enzymatic activities section from the manuscript. We did necessary revision regarding this throughout the MS.

Reviewer's comment:

Materials and methods

Comment 8: Line 519: Did the authors extract DNA from the digesta of hindgut? Please mention.

Our response:

Thanks for the comment. Yes, we extracted DNA from the digesta of hindgut. We have revised the information in the MS.

Reviewer's comment:

Comment 9: Line 560 and Line 608: Same sub-topic. Please use more specific sub-topics.

Our response:

We have added specific sub-topic.

Reviewer's comment:

Comment 10: Line 583: The authors have used the same abbreviations (i.e., T1, T2 and T3) to denote three other diets in Line 482.

Our response:

Thanks for the comments. To avoid confusion, we have revised the treatment names in growth performance (Feeding Trial) experiments (F-T1, F-T2, F-T3)

Reviewer's comment:

Tables

Comment 11: Please provide footnotes for abbreviations used in the tables.

Our response:

We have provided footnotes for abbreviations used in the tables.

Reviewer's comment:

Comment 12: Table 5: Would it not be better to use a nested ANOVA to analyze the results in this table? If you considered this but decided not to, what were your reasons?

Our response:

We have used nested NOVA and presented the data in a table accordingly.

Thank you respected reviewer for giving your valuable time. Hopefully, you will find our revision satisfactory.

Reviewer #2

Reviewer's comment:

Gram-positive *Enterococcus faecalis* (Ef) belongs to the lactic acid bacteria (LAB) group. Ef is formerly classified as part of the group D streptococcus system. These bacteria is found in a diverse number of environments such as the gastrointestinal tract of healthy humans. However, it is a opportunistic pathogen in humans, causing even life-threatening infections, especially in hospital environments, were naturally high levels of antibiotic resistance Ef contribute to pathogenicity. In aquaculture nutrition field, it is well known that Ef, was isolated from the intestines of healthy humans and demonstrated Immune stimulation properties and adjuvant benefits when administered in a lysed or inactivated form.

It is well known that Nile tilapia, is the most important aquacultured species in the tropical regions of the planet. Although, tilapia tolerate a wide gamma of adverse water quality and external stressors better than most commercial aquaculture species, it is evident that this species has certain susceptibility to specific pathogens. These pathogens produce a diverse number of diseases. Such as Streptococcosis, caused by *Streptococcus iniae* and *S. galactidae* or Septicemia caused by *Aeromonas hydrophila* or *Trichodina*, produced by a protozoan parasite or *Columnaris* caused by the myxobacteria, *Flexibacter columnaris*.

Although this manuscript presents results derived from a high quality research (novelty) and well designed study, there are some relevant concerns related to experimental design. Among these concerns, surge the question of why authors did select *Enterococcus faecalis* as pathogen for this study? If there are other more recurrent pathogens affecting Nile tilapia?. To this respect, reviewer strongly suggest that authors should include (in introduction), a brief paragraph talking about how *E. faecalis* infect Nile tilapia cultures in Bangladesh. I mean, how this pathogen agent is able to reach rearing water of farms, since *E. faecalis* is per sé originated in human intestines hence present in urban waste waters.

Our response:

Thank you so much for making these precious comments about the bacterium, *Enterococcus faecalis*. Yes, it is true that in aquaculture nutrition field, many strains of *E. faecalis* isolated from the intestines of healthy humans as well as healthy fish have already been established as probiotic bacteria to enhance the growth and immune response of many fish species globally. However, our research group identified *E. faecalis* as a pathogen of Nile tilapia with high mortality during the year 2015-2017 and published the findings in *Scientific Reports*

(Rahman et al., 2017). Our further metagenomics and *in vivo* studies confirmed the involvement of *Enterococcus* species in streptococcosis of Nile tilapia in Bangladesh (Akter et al., 2021). We also performed whole-genome sequencing (WGS) of several fish pathogenic *E. faecalis* strains isolated from streptococcosis of Nile tilapia. The WGS data of two fish pathogenic *E. faecalis* have been deposited in the NCBI Gene Bank under accession numbers CP046022 and JADBGH010000000. After genomic analysis, we found that these strains conserved 69 virulence factors including genes associated with gelatinase toxin and capsule producing genes for their survival. Furthermore, 39 antibiotic-resistance genes against sixteen groups of antibiotics were identified in the genome sequences of the strains. All of the strains conserved multiple antibiotic-resistance genes including macrolide-lincosamide-streptogramin (*MLS*) resistant genes, peptide antibiotic-resistance genes (*LiaR*, *LiaS*, *LiaF*, *MprF*, *GdpD*, *PgsA*, *rpoB* and *rpoC*), isoniazid and triclosan resistant genes, diaminopyrimidine resistant genes, aminoglycosides conferring genes, cycloserine resistance genes, and fluoroquinolones and quinolones resistance genes.

Though *Streptococcus iniae*, *S. agalactiae*, *S. parauberis*, and *S. dysgalactiae* are recognized as the prominent causative agents for streptococcosis in different geographical regions. However, *Lactococcus*, *Vagococcus*, and *Enterococcus* are also isolated from the diseased fish (El-Sayed and El-Gheit, 2005; Osman et al., 2017; Mishra et al., 2018). Due to inadequate diagnostic facilities at fish farms, the majority of causative agents responsible for diseases remained uncharacterized. Recent advancements in omics technologies, especially in next generation sequencing and downstream bioinformatics analysis facilitated more precise identification and in-depth analysis of microbial communities in a given sample as well as involvement of pathogens in a disease. In more recent times, *E. faecalis* is reported as notable causative agents for streptococcosis (Akter et al., 2021; Ehsan et al., 2021; Osman et al., 2017). This bacterium is posing a constant threat to fisheries and aquaculture in Asian countries such as India (Arumugam et al., 2017), Bangladesh (Rahman et al., 2017; Akter et al., 2020; Akter et al., 2021; Ehsan et al., 2021), Philippines (Reyes et al., 2020), Indonesia (Rizkiantino et al., 2020) as well as some African countries like Egypt (Zahran et al., 2019). Due to the existing state of sewage contamination in fish farms, where infections are rampant, the bacteria offer considerable health concerns to both animals and humans (Wang et al., 2014).

We have added the necessary information about this pathogen in the introduction.

Supportive References are listed below-

- Arumugam U, Stalin N, Rebecca GP. 2017. Isolation, molecular identification and antibiotic resistance of *Enterococcus faecalis* from diseased Tilapia. Int J Curr Microbiol Appl Sci 6:136–146. <https://doi.org/10.20546/ijcmas.2017.606.016>
- Rahman M, Rahman M, Deb SC, Alam M, Islam M. 2017. Molecular identification of multiple antibiotic resistant fish pathogenic *Enterococcus faecalis* and their control by medicinal herbs. Sci Rep 7:1-11. <https://doi.org/10.1038/s41598-017-03673-1>
- Akter T, Rahman MM, Tay AC, Ehsan R, Islam MT. 2020. Whole-Genome Sequence of Fish-Pathogenic *Enterococcus faecalis* Strain BFFF11. Microbiology Resource Announcements. 13;9(7):e01447-19. <https://doi.org/10.1128/MRA.01447-19>
- Akter T, Foysal MJ, Alam M, Ehsan R, Paul SI, Momtaz F, Siddik MA, Tay AC, Fotedar R, Gupta SK, Islam T. 2021. Involvement of *Enterococcus* species in streptococcosis of Nile tilapia in Bangladesh. Aquaculture 531:735790. <https://doi.org/10.1016/j.aquaculture.2020.735790>
- El-Sayed, N., El-Gheit, A., 2005. Streptococcus infections among farmed and wild tilapia fish in Egypt. The Egyptian Journal of Experimental Biology. 1, 1–7.
- Mishra, A., Nam, G.H., Gim, J.A., Lee, H.E., Jo, A., Kim, H.S., 2018. Current challenges of *Streptococcus* infection and effective molecular, cellular, and environmental control methods in aquaculture. Molecules and Cells 41 (6), 495–505. <https://doi.org/10.14348/molcells.2018.2154>
- Ehsan R, Alam M, Akter T, Paul SI, Foysal MJ, Gupta DR, Islam T, Rahman MM. 2021. *Enterococcus faecalis* involved in streptococcosis like infection in silver barb (*Barbonymus gonionotus*). Aquac Rep 21:100868. <https://doi.org/10.1016/j.aqrep.2021.100868>
- Osman KM, Al-Maary KS, Mubarak AS, Dawoud TM, Moussa IM, Ibrahim MD, Hessain AM, Orabi A, Fawzy NM. 2017. Characterization and susceptibility of streptococci and enterococci isolated from Nile tilapia (*Oreochromis niloticus*) showing septicemia in aquaculture and wild sites in Egypt. BMC Vet Res 13:357 <https://doi.org/10.1186/s12917-017-1289-8>
- Reyes AT, Madrid FGN. 2020. Isolation and biochemical identification of *Enterococcus faecalis* on moribund Nile Tilapia (*Oreochromis niloticus* L). Adv Phar Clin Tria 5:000177. <https://doi.org/10.23880/apct-16000175>
- Rizkiantino R, Wibawan IW, Pasaribu FH, Soejoedono RD, Arnafia W, Ulyama V, Wibowo DB. 2020. Isolation and characterization of the *Enterococcus faecalis* strain isolated

from red tilapia (*Oreochromis hybrid*) in Indonesia: A preliminary report. Survey Fish Sci 7:27-42. <http://dx.doi.org/10.18331/SFS2020.7.1.3>

Zahran E, Mahgoub HA, Abdelhamid F, Sadeyen JR, Risha E. 2019. Experimental pathogenesis and host immune responses of *Enterococcus faecalis* infection in Nile tilapia (*Oreochromis niloticus*). Aquaculture 512:734319. <https://doi.org/10.1016/j.aquaculture.2019.734319>

Wang S, Hibberd ML, Pettersson S, Lee YK. *Enterococcus faecalis* from healthy infants modulates inflammation through MAPK signaling pathways. PloS one. 2014 May 15;9(5):e97523. <https://doi.org/10.1371/journal.pone.0097523>

Reviewer's comment:

The other big concern is related to probiotic microorganisms used in present study. Although *Bacillus haynesii* strain CD223 and *Advenella mimigardefordensis* strain SM421 were isolated from marine soil, they were used as probiotic agents in a freshwater species. In reviewer's opinion, probiotic - prebiotic research, should be targeted to candidate microorganisms highly available in similar environments where the host is reared.

Our response:

Thank you so much for the valuable comments. Marine environment represents a promising source of new, unconventional bioactive compounds with health-promoting abilities, which can be used as food supplements in freshwater fish species Nile tilapia (Makled et al., 2017). Even though, to the best of the author's knowledge, results from numerous applied aquaculture research trials have effectively confirmed that the use of marine bacterial species isolated from marine sediments, algae, sponge and corals can improve the farming environment and used as feed supplements in freshwater fish species (Makled et al., 2017; Makled et al., 2019; Makled et al., 2020; Hamdan et al., 2016; Soltani et al., 2019; Paul et al., 2021; Rahman et al., 2020; Paul and Rahman, 2022).

Our potential beneficial marine bacteria *Bacillus haynesii* strain CD223 and *Advenella mimigardefordensis* strain SM421 did not show any cytotoxic effects, and both of the strain can grow both in freshwater nutrient broth and marine broth. In *in vivo*, both of the strains developed disease resistance in tilapia and did not show any adverse effects on fish behavior and hematological parameters. Tilapia are euryhaline species, and are able to live and reproduce in a wide range of salinities. Application of these marine bacterial strains (CD223 and SM421) also enhanced the growth performances significantly and developed healthy gut

bacteriome communities in fish. Therefore, based on the above-mentioned facts the authors think that these two potential marine bacterial strains could be a significant addition to the fish health management research. However, in respect to your valuable opinion, we have replaced the word “probiotic” by “beneficial marine bacteria” in the revised MS.

Supportive References are listed below-

- Makled SO, Hamdan AM, El-Sayed AF, Hafez EE. 2017. Evaluation of marine psychrophile, *Psychrobacter namhaensis* SO89, as a probiotic in Nile tilapia (*Oreochromis niloticus*) diets. Fish & shellfish immunology. 61:194-200. <https://doi.org/10.1016/j.fsi.2017.01.001>
- Hamdan AM, El-Sayed AF, Mahmoud MM. 2016. Effects of a novel marine probiotic, *Lactobacillus plantarum* AH 78, on growth performance and immune response of Nile tilapia (*Oreochromis niloticus*). Journal of applied microbiology. 120(4):1061-73. <https://doi.org/10.1111/jam.13081>
- Makled SO, Hamdan AM, El-Sayed AF. 2019. Effects of dietary supplementation of a marine thermotolerant bacterium, *Bacillus paralicheniformis* SO-1, on growth performance and immune responses of Nile tilapia, *Oreochromis niloticus*. Aquaculture Nutrition. 25(4):817-27. <https://doi.org/10.1111/anu.12899>
- Makled SO, Hamdan AM, El-Sayed AF. 2020. Growth promotion and immune stimulation in Nile tilapia, *Oreochromis niloticus*, fingerlings following dietary administration of a novel marine probiotic, *Psychrobacter maritimus* S. Probiotics and antimicrobial proteins. 12(2):365-74. <https://doi.org/10.1007/s12602-019-09575-0>
- Soltani M, Ghosh K, Hoseinifar SH, Kumar V, Lymbery AJ, Roy S, Ringø E. 2019. Genus *Bacillus*, promising probiotics in aquaculture: aquatic animal origin, bio-active components, bioremediation and efficacy in fish and shellfish. Reviews in Fisheries Science & Aquaculture. 27(3):331-79. <https://doi.org/10.1080/23308249.2019.1597010>
- Paul SI, Rahman MM, Salam MA, Khan MAR, Islam MT. 2021. Identification of marine sponge-associated bacteria of the Saint Martin's island of the Bay of Bengal emphasizing on the prevention of motile *Aeromonas* septicemia in *Labeo rohita*. Aquaculture 545:737156. <https://doi.org/10.1016/j.aquaculture.2021.737156>
- Rahman MM, Paul SI, Akter T, Tay AC, Foysal MJ, Islam MT. 2020. Whole-genome sequence of *Bacillus subtilis* WS1A, a promising fish probiotic strain isolated from

marine sponge of the Bay of Bengal. Microbiology Resource Announcements. 9(39):e00641-20. <https://doi.org/10.1128/MRA.00641-20>

Paul SI, Rahman MM. Draft Genome Sequence of *Bacillus subtilis* YBS29, a Potential Fish Probiotic That Prevents Motile *Aeromonas* Septicemia in *Labeo rohita*. 2022. Microbiology Resource Announcements.e00915-22. <https://doi.org/10.1128/mra.00915-22>

Reviewer's comment:

On the other hand, it is strongly recommended all along the manuscript, *Enterococcus faecalis* should be mentioned only at the beginning. Subsequently, it can be mentioned as E. faecalis.

Our response:

We have revised it where necessary.

Reviewer's comment:

Finally, in the manuscript, M&M and discussion are misplaced. Results are misleading since in some parts, it seems that are mixed with M&M.

Our response:

Thank you so much for your valuable comments. We have revised the manuscript according to the reviewer's comments, corrected the spelling and deleted the repetitive parts.

Thank you so much respected reviewer for giving your valuable time. Hopefully, you will find the revised manuscript satisfactory.

October 31, 2022

Prof. Md. Mahbubur Rahman
Bangabandhu Sheikh Mujibur Rahman Agricultural University
Institute of Biotechnology and Genetic Engineering
Salna
Gazipur 1706
Bangladesh

Re: Spectrum02542-22R1 (Suppression of Streptococcosis and Modulation of the Gut Bacteriome in Nile Tilapia (*Oreochromis niloticus*) by Marine Sediment Bacteria *Bacillus haynesii* and *Advenella mimigardefordensis*)

Dear Prof. Md. Mahbubur Rahman:

Your manuscript has been accepted, and I am forwarding it to the ASM Journals Department for publication. You will be notified when your proofs are ready to be viewed.

Sincerely,

Artem Rogovskyy
Editor, Microbiology Spectrum
